# High-Precision Wafer Bonding Alignment Mark Using Moiré Fringes and Digital Grating

**DOI:** 10.3390/mi13122159

**Published:** 2022-12-07

**Authors:** Jianhan Fan, Sen Lu, Jianxiao Zou, Kaiming Yang, Yu Zhu, Kaiji Liao

**Affiliations:** 1School of Automation Engineering, University of Electronic Science and Technology of China, Chengdu 611731, China; 2State Key Laboratory of Tribology, Department of Mechanical Engineering, Tsinghua University, Beijing 100084, China; 3Shenzhen Institute for Advance Study, University of Electronic Science and Technology of China, Shenzhen 518110, China; 4State Grid Sichuan Electric Power Corporation Metering Center, Wanjing Road I, Chengdu 610000, China

**Keywords:** wafer bonding, moiré fringe, wafer bonding alignment

## Abstract

This paper investigates a moiré-based mark for high-precision wafer bonding alignment. During alignment, the mark is combined with digital grating, which has the benefits of high precision and small size. A digital grating is superimposed on the mark to generate moiré fringes. By performing a phase calculation on the moiré fringe images corresponding to the upper and lower wafers, the relative offset of the upper and lower wafers can be accurately calculated. These moiré fringes are exceptionally stable, thereby enhancing the alignment stability. In this study, through practical experiments, we tested the rationality and practicability of the mark.

## 1. Introduction

Moore’s law predicts that the scale of integrated circuits in chips will become larger and more prominent in the future, but reducing the size of transistors to improve performance is quite challenging. Accordingly, three-dimensional (3D) integration in integrated circuit technology has emerged as a promising new direction [1,2]. Wafer bonding technology is an essential technology for achieving precise 3D integration. Wafer bonding is the process of bonding two silicon-based wafers with circuits etched together as shown in Figure 1. In the bonding phase, ensuring the two wafers are aligned is necessary.

The alignment process in wafer bonding is one of the most critical steps in the wafer bonding process, and the alignment accuracy directly determines the quality of the bonding. To achieve high alignment accuracy, the relative displacement of the upper and lower wafers in the XY plane after bonding should be less than ±50 nm, and the relative deflection of the upper and lower wafers should be less than ±1 μrad. For such precision, a high-precision alignment technique is necessary.

Wafer bonding alignment methods are generally divided into designing mechanical structures and methods based on alignment mark recognition. Sang Hwui Lee et al. designed a keyed alignment structure to facilitate alignment [3]. Alignment was performed by adding a PECVD oxide layer to the etched wafer to obtain a trapezoidal structure. Isao Sugaya et al. proposed using a multi-axis interferometer and a weight sensor to make a special clipper for alignment [4]. Chenxi Wang et al. designed a centrosymmetric grating mark combined with image processing for alignment [5]. Using the designed mechanical structure yielded low alignment accuracy and was difficult to achieve and expensive. Designing alignment marks with digital image processing methods for alignment is well recognized and widely used for causing less damage to the wafer, lower production costs, and facilitating implementation.

Today, the cross mark is the most commonly used alignment mark in the wafer bonding alignment process. Typically, a direct image algorithm is used to obtain the center point position coordinates of the child and mother marks. Then, using two position coordinates can calculate the deviation of the upper and lower wafers in the alignment process. Calculating the mark’s center position employs direct digital image processing, and the calculated center position coordinate error is too large. Therefore, meeting the precision requirements of the front-end wafer bonding alignment process is challenging. Chenxi Wan and Boyang Huang proposed a moiré-fringe-based centrosymmetric square mark. The mark can accurately calculate the deviation of the upper and lower wafers in the alignment process. Nonetheless, this mark has several deficiencies: controlling the distance between the upper and lower wafers is required; otherwise, diffraction effects will occur, resulting in poor image quality; additionally, it is too large and will occupy a portion of the lithography space [6,7,8].

Nevertheless, the detection method based on moiré fringes still has excellent development potential. Moiré fringes generated by two similar linear gratings or concentric circular gratings are frequently used to measure displacement because they can more easily reveal minute relative displacements. For moiré fringes to be utilized in the design of wafer bonding alignment marks, the measurement of the alignment deviation of the upper and lower wafers should be equivalent to the measurement of the relative displacement of the upper and lower wafers.

Based on the aforementioned concepts and requirements, a novel two-dimensional centrosymmetric mark for non-destructive measurement of the alignment deviation of the upper and lower wafers during the wafer bonding process was designed, and the alignment process was also optimized. It has the following advantages: (1) high measurement accuracy; (2) small mark; (3) capability of measuring multiple degrees of freedom concurrently; (4) capability of measuring significant displacement differences, etc.

## 2. Alignment Process

The design of the alignment method must be compatible with the bonding process, so the alignment mark must be based on the actual situation of the existing bonding equipment. The alignment process and the structure of the bonding equipment in the wafer bonding process are extremely complex, and many of the steps involve improving motion precision, which is not the focus of this article. For the purpose of this study’s subject matter, the alignment procedure can be simplified as follows:

### 2.1. Coaxial Alignment

Align the four cameras, aligning the upper and lower cameras on both sides (with the same field of view) and the left and right sides (the range of the field of view in the Y direction is the same). When alignment is complete, each camera moves from its coaxial position to its alignment position. The bonding equipment is as shown in Figure 2.

### 2.2. Lower Wafer Position Record

The lower carrier stage (which carries the lower wafer, with the bonding surface facing up) moves from the pick-up position to the alignment position. The upper camera captures the mark of the lower wafer and calculates and records the center coordinates of the mark. On the carrier stage (adsorb the upper wafer; the bonding side is facing down), stay in the pick-up position, as shown in Figure 3a.

### 2.3. Upper Wafer Position Record

The lower carrier stage moves to the pick-up position, the upper carrier stage moves to the alignment position, the lower camera captures the upper wafer, and the mark center coordinates are calculated and recorded, as shown in Figure 3b.

### 2.4. Position Deviation Correction

The lower wafer travels to the coordinate position recorded by the lower wafer in the alignment position. The motion system calculates the deviation between the two wafers in the alignment position and corrects the deviation.

The structure of the bonding equipment is complex, and it has strict requirements for size, vibration, precision, cost, etc. It is difficult to alter the motion process significantly, and only a portion of it can be optimized. Therefore, the design of the alignment method must account for the movement process, and it is necessary to ensure the alignment’s precision without modifying or optimizing the movement process.

## 3. Mark Design

### 3.1. Moiré Fringe Formation Principle

Moiré fringes were initially identified as a phenomenon observed when two similar but not identical grating systems are overlapped and reflected or passed through one another. In addition to the original grating, a periodic structure can be observed at this point. The phenomenon of grating and shadow overlapping to form moiré fringes emerges with industrial growth, but its essence is consistent with the formation principle of moiré fringes. Consequently, the general moiré fringes are described as follows: moiré is the result of the interaction between two or more co-located periodic structures; it is a spatially modulated intensity pattern created by alternating dark and bright stripes; these dark stripes constitute the moiré pattern and are referred to as moiré [9,10,11,12].

In recent years, moiré fringes have been widely used to measure displacement, deformation, and other fields [12,13,14,15]. In this paper, calculating the deviation of the upper and lower wafers corresponds to calculating the relative displacement of the upper and lower wafers. Hence, the moiré fringes are ideally suited to this situation.

### 3.2. Calculation of Moiré Fringes

Using the overlapping of two gratings as an example (Figure 4), the periods and angles of the two gratings are $P_{1}\;$ and $P_{2}$, respectively, and the angle between the two gratings is θ. Typically, the difference between $P_{1}$ and $P_{2}$ is minimal, and the wavelength of the incident light is considerably shorter than the grating period. The effect of diffraction is also negligible. Currently, the period W of the formed moiré fringes is as follows:(1)$$W=\frac{P_{1}P_{2}}{\sqrt{P_{1}^{2}+P_{2}^{2}-2P_{1}P_{2}cos\theta }}$$

When moiré fringes are used for measuring displacement, θ is typically set close to zero, and $W=P_{1}P_{2}/\left|P_{1}-P_{2}\right|$, where $P_{1}$ is the grating to be measured and $P_{2}$ is the detection grating. The newly formed moiré fringes have a magnification effect in relation to the grating being measured, with the magnification formula being $K=W/P_{1}=P_{2}/\left|P_{1}-P_{2}\right|$. Clearly, as the magnification increases, the periods of the two gratings become closer together.

### 3.3. Synthesis of Moiré Fringes

Digital grating is an artificially produced digital image of grating type, and its parameters—including shape, period, and duty cycle—can be artificially defined. The digital grating operates similarly to a digital filter, and a matrix of B=m×n represents its mathematical model; m and n represent the number of rows and columns of the image resolution obtained by the imaging system, respectively, and the matrix contains only 0 and 1 elements. When the element in B is 0, the digital grating is “opaque”, whereas when it is 1, the digital grating is “transmitting”. Then, the digital raster superimposes the actual raster to create a new image:(2)$$C_{1}=A\times B$$
(3)$$C_{2}=A\times \bar{B}$$

A=m×n is the actual grating image, while $\bar{B}=1-B$ is the reverse. The superimposed image is as shown in Figure 5.

The line width of the digital grating is i, and the duty cycle is 50%. The two overlapping grating images are then integrated to produce two light intensity curves $I_{1}$ and $I_{2}$:(4)$$I_{1j}=\int_{x=1}^{n}\int_{y=1+2i\left(j-1\right)}^{i+2i\left(j-1\right)}I\left(x,y\right)dxdy$$
(5)$$I_{1}=\left[I_{11},I_{12},\ldots ,I_{1m}\right]$$
(6)$$I_{2j}=\int_{x=1}^{n}\int_{y=1+2i\left(j-1\right)}^{2i+2i\left(j-1\right)}I\left(x,y\right)dxdy$$
(7)$$I_{2}=\left[I_{21},I_{22},\ldots ,I_{2m}\right]$$

The moiré signal curve I can then be calculated as follows:(8)$$I=\frac{I_{1}-I_{2}}{I_{1}+I_{2}}$$

The extracted light intensity curve and moiré signal curve are shown in Figure 6.

### 3.4. Solution of Grating Displacement

The ideal state of the moiré signal is a triangular wave. However, in practice, due to the limitations of camera imaging function, environmental interference, the precision of grating fabrication, etc., the moiré signal (I) is typically a sine wave. Therefore, the extracted moiré signal (I) can be considered a sinusoidal signal with interference. Using the variational mode decomposition (VMD) algorithm [16], the interference signal in the moiré signal (I) can be removed, and the fundamental signal v(n) can be obtained, as depicted in Figure 7.

Then, we obtain the phase difference between the two moiré signals. The discrete Fourier transform (DFT) algorithm transforms two discrete signals with signal length *N* into the following form [17]:(9)$$V_{i}\left(k\right)=\overset{\infty }{\underset{n=-\infty }{\sum }}v_{i}\left(n\right)e^{\frac{-j2\pi kn}{N}}=\overset{N-1}{\underset{n=0}{\sum }}v_{i}\left(n\right)e^{\frac{-j2\pi kn}{N}},i=1,2$$

According to Euler’s formula
(10)$$e^{\frac{-j2\pi kn}{N}}=cos\left(\frac{2\pi kn}{N}\right)-jsin\left(\frac{2\pi kn}{N}\right)$$
there are
(11)$$V_{i}\left(k\right)=\overset{N-1}{\underset{n=0}{\sum }}v_{i}\left(n\right)\left(cos\left(\frac{2\pi kn}{N}\right)-jsin\left(\frac{2\pi kn}{N}\right)\right),i=1,2$$

The phases of the two discrete signals are as follows:(12)$$\varphi_{i}=arctg\frac{Im\left(V_{i}\left(k\right)\right)}{Re\left(V_{i}\left(k\right)\right)}=\frac{\sum_{n=0}^{N-1}v_{i}\left(n\right)sin\left(\frac{2\pi }{N}nk\right)}{\sum_{n=0}^{N-1}v_{i}\left(n\right)cos\left(\frac{2\pi }{N}nk\right)},i=1,2$$

The final phase difference between the two signals is as follows:(13)$$\Delta \varphi =\varphi_{1}-\varphi_{2}$$

The displacement difference between the two grating images can be obtained by using the phase difference:(14)$$\Delta D=\frac{\Delta \varphi }{2\pi }L$$
where *L* is the grating’s period. The deviation between the upper and lower wafers in the XY plane is equivalent to the relative displacement between the upper and lower wafers in the XY plane. Thus, the relative displacement between the gratings can be precisely determined, as well as the precise deviation between the upper and lower wafers.

### 3.5. Design Results

Our bonding device’s imaging system parameters are as follows: the camera’s resolution is 3856 × 2764, the pixel size is 1.67 μm × 1.67 μm, and the objective lens’s theoretical magnification is 5. Consequently, one pixel in the image captured by the imaging system should be calibrated as a rectangular block measuring 0.334 μm × 0.334 μm in the actual space. However, lens processing, camera damage, environmental interference, and other factors will result in a theoretical calibration result that differs from the actual one. To improve the accuracy of subsequent calculations, we calibrated the imaging system using Zhang’s calibration method [18]: in actual space, one pixel corresponds to a 0.360 μm × 0.360 μm rectangular square.

The following mark was designed by combining the above theory with the actual working conditions of the bonding equipment and the imaging system (Figure 8):

The intended marking period is 6 μm, and the duty cycle is 50%; consequently, the line width is 3 μm, the number of cycles is 11, and the total mark size is 63 μm×63 μm. Create a cross mark in the mark’s center, which has two advantages: (1) the NCC algorithm [19] can be used to quickly locate the approximate position of the mark through this cross mark, and the mark can be intercepted, which significantly reduces the amount of calculation; (2) the relative displacement of the two intercepted marks can be calculated, replacing a large-scale displacement difference with a small-scale displacement difference, and the solution result must be within one pixel (the NCC positioning accuracy is pixel-level). There is no need to consider the relative displacement as being excessively large, causing inaccurate results. The solution accuracy is enhanced, and the calculation difficulty is diminished.

Except for the cross marks, the remaining components are plane gratings, allowing the X direction, Y direction, and relative deflection of the upper and lower wafers to be measured by superimposing different digital gratings in a single image.

Extract the calculation area of two gratings, as shown in Figure 9a, and the relative displacement of the two gratings in the X direction can be determined. As depicted in Figure 9b, by extracting the calculation area of the two gratings, the relative displacement of the two gratings along the *Y*-axis can be determined. By extracting the two calculation regions of a grating, as depicted in Figure 9c, the trigonometric function can be used to calculate the grating’s horizontal deflection. The relative deflection of the two gratings can be calculated by combining them in the horizontal direction.

According to the above design, the bonding device’s upper two cameras or lower two cameras can be removed, and the infrared light path can be increased (Figure 10).

As shown in Figure 11, the alignment process can be enhanced based on the improved structure.

Through this enhancement, the coaxial alignment process in the initial alignment process can be eliminated, the coaxial alignment error can be directly eliminated, and the number of cameras and associated costs can be decreased. After implementing the moiré fringe solution, the designed mark is smaller than the original mark, and the deviation calculation accuracy of the upper and lower wafers is improved. Compared to the original direct digital image processing, the moiré-fringe-based solution method requires less computation, operates more quickly, and provides greater precision.

## 4. Experimental Verification

In order to test the practicality of the markers mentioned above, we designed and constructed the following experimental equipment (Figure 12):

The experimental design of this paper is as follows: whenever the micro-movement stage moves, the capacitive sensor records the displacement of the micro-movement stage while, simultaneously, the camera captures the position of the grating, and this is repeated multiple times. The capacitive sensor’s measurements are used as the actual displacement. The results of the displacement calculation method utilized in this paper were compared to the displacement results calculated by widely recognized image calculation software available on the market. The outcomes are shown in Figure 13 and Figure 14.

Since the capacitive sensor’s precision is exceptionally high, the measured parameters are used as the actual displacement. As calculated in this paper, the average error between the X direction and the capacitive sensor was 5.22 nm. In contrast, the average error between the X direction and capacitive sensor calculated using commercial software was 48.54 nm. The presented methodology calculated an average error of 8.26 nm between the Y direction and the capacitive sensor, whereas commercial software calculated an average error of 37.13 nm. The presented methodology had a repeat calculation error of ±3.23 nm in the X direction and a repeat calculation error of ±2.87 nm in the Y direction. Clearly, the calculation accuracy was greatly improved after using the grating designed in this paper and the digital grating. The errors in the X and Y directions were within 10 nm, significantly improving the alignment process’s accuracy.

## 5. Conclusions

Using grating marks and digital gratings based on moiré fringes can significantly improve the accuracy of calculating the deviation of the upper and lower wafers in the alignment process, and it can control the estimated error to within 10 nm. These technical achievements satisfy all requirements for high-precision bonding. Moreover, the calculation effect and speed are superior to those of commercially available image processing software. In addition, the designed grating marks are small enough to increase the space on the wafer for etched circuits. This marker can also continuously optimize the moiré signal by adjusting the superimposed digital grating to achieve the best solution results and can simultaneously calculate the results for all three degrees of freedom. It is also possible to cut down on the number of keys and processes, which not only lowers the cost but also cuts out a part of the bonding process, which lowers the overall error rate.

## Figures and Tables

**Figure 1 micromachines-13-02159-f001:**
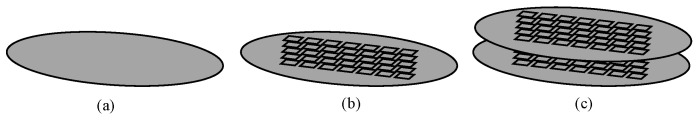
Wafer bonding schematic: (**a**) silicon-based wafer; (**b**) wafer with circuits; (**c**) wafer pair bonding.

**Figure 2 micromachines-13-02159-f002:**
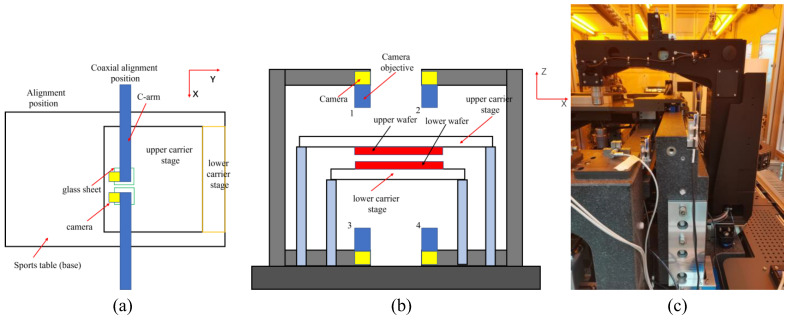
Bonding equipment: (**a**) simple structure of bonding equipment (coaxial alignment process) (top view); (**b**) bonding equipment schematic diagram (front view); (**c**) wafer bonding alignment machine.

**Figure 3 micromachines-13-02159-f003:**
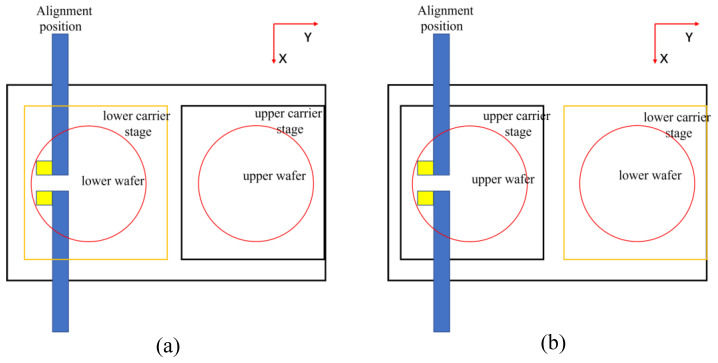
Alignment process: (**a**) lower wafer position record; (**b**) upper wafer position record.

**Figure 4 micromachines-13-02159-f004:**
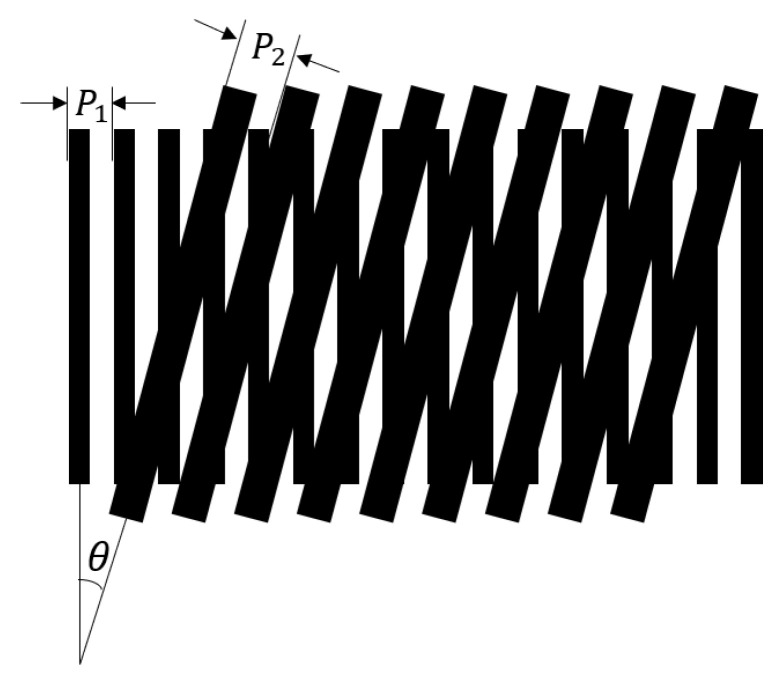
Formation of moiré fringes.

**Figure 5 micromachines-13-02159-f005:**
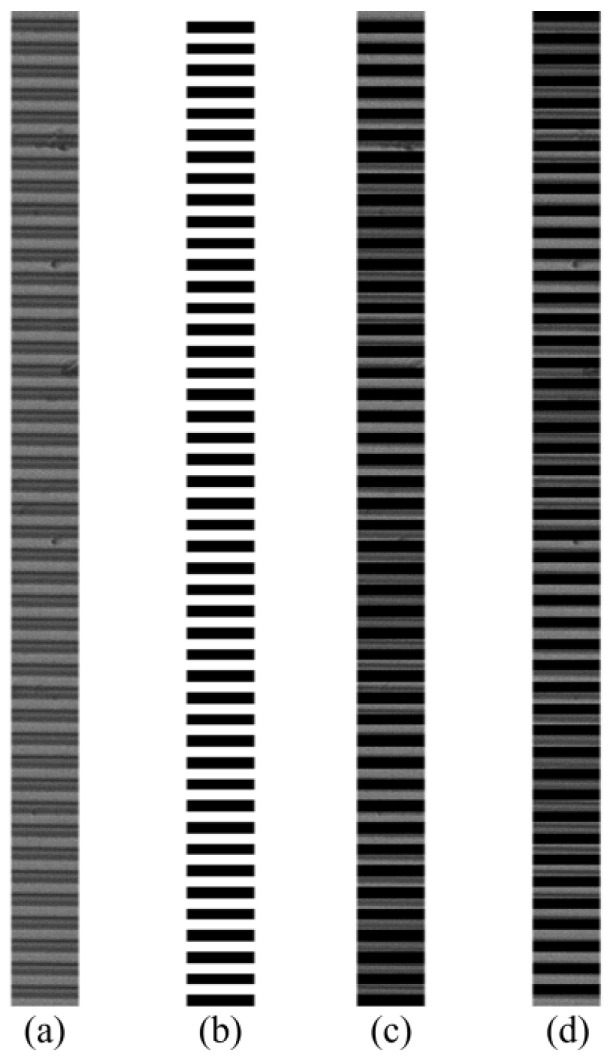
(**a**) Actual grating; (**b**) digital grating; (**c**) $C_{1}$; (**d**) $C_{2}$.

**Figure 6 micromachines-13-02159-f006:**
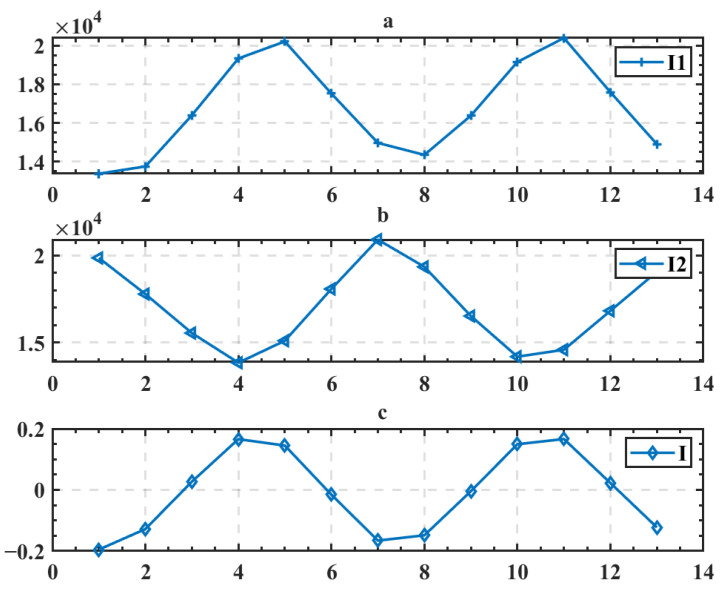
(**a**) Light intensity curve $I_{1}$; (**b**) light intensity curve $I_{2}$; (**c**) moiré signal I.

**Figure 7 micromachines-13-02159-f007:**
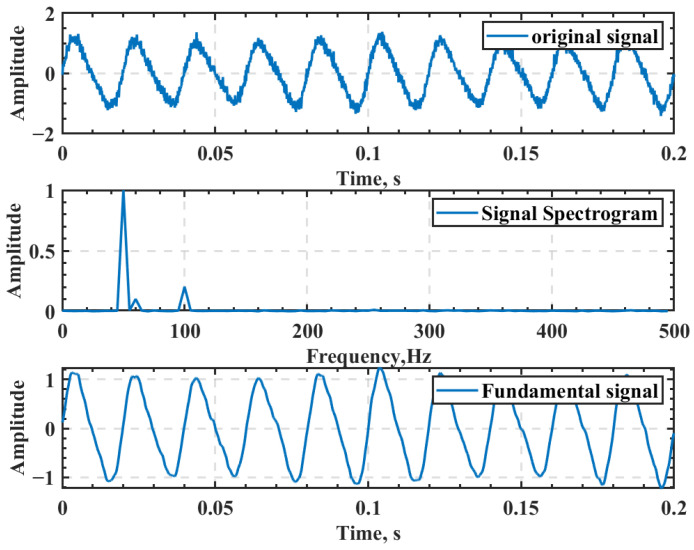
VMD filter effect.

**Figure 8 micromachines-13-02159-f008:**
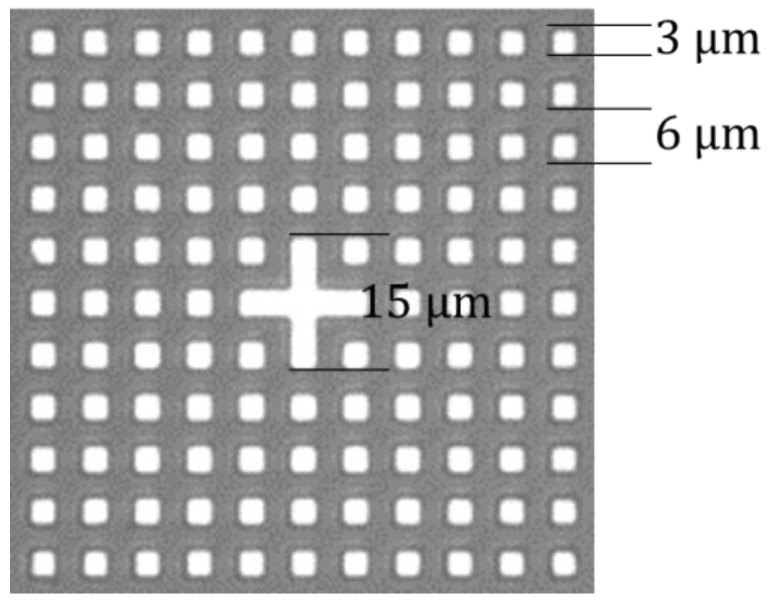
Design mark.

**Figure 9 micromachines-13-02159-f009:**
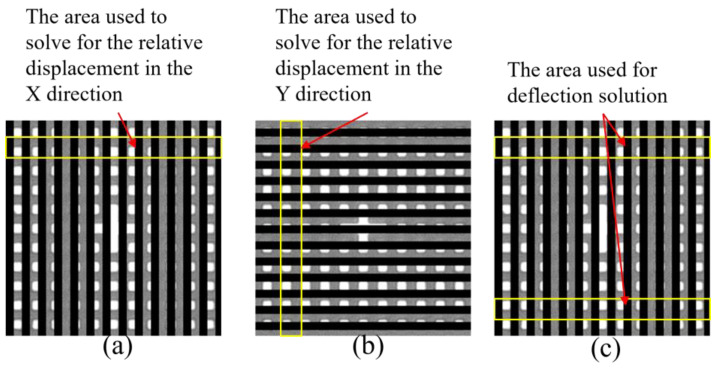
(**a**) Relative displacement calculation along the X-axis; (**b**) the relative displacement calculation in the Y direction; (**c**) the relative deflection calculation of the two gratings.

**Figure 10 micromachines-13-02159-f010:**
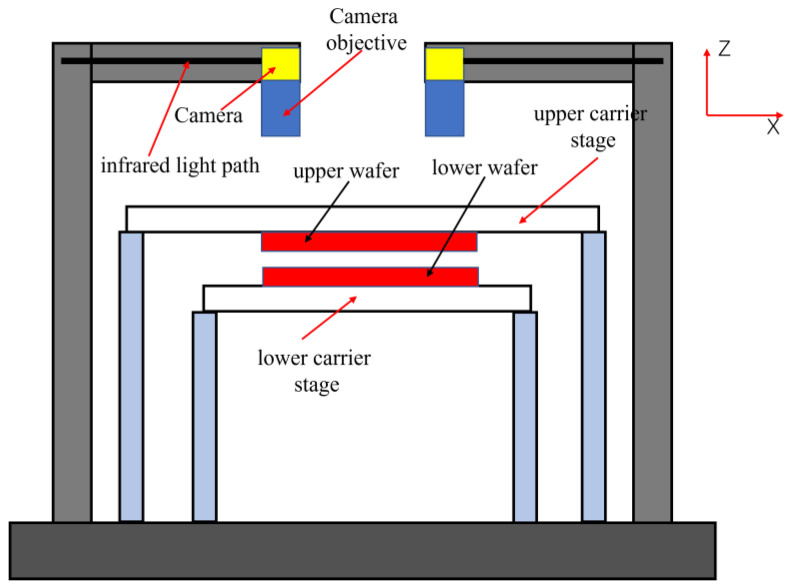
Improved structure.

**Figure 11 micromachines-13-02159-f011:**
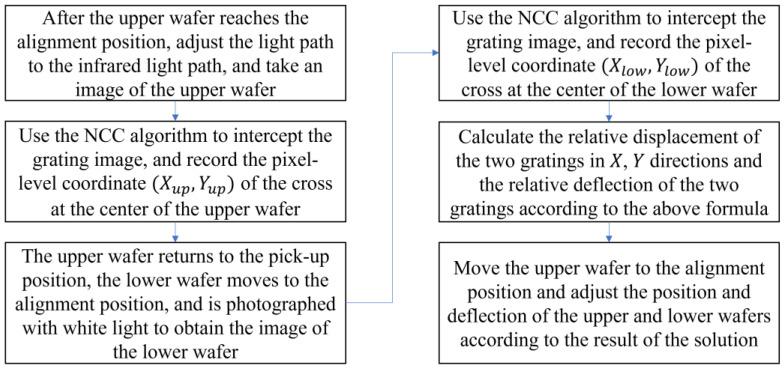
Improved alignment process.

**Figure 12 micromachines-13-02159-f012:**
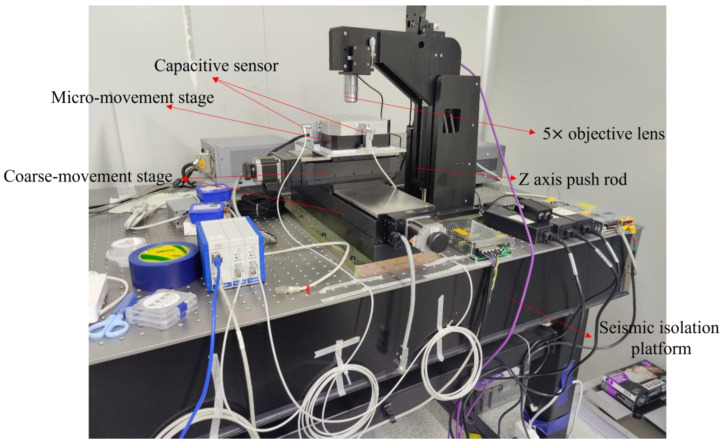
Experimental equipment.

**Figure 13 micromachines-13-02159-f013:**
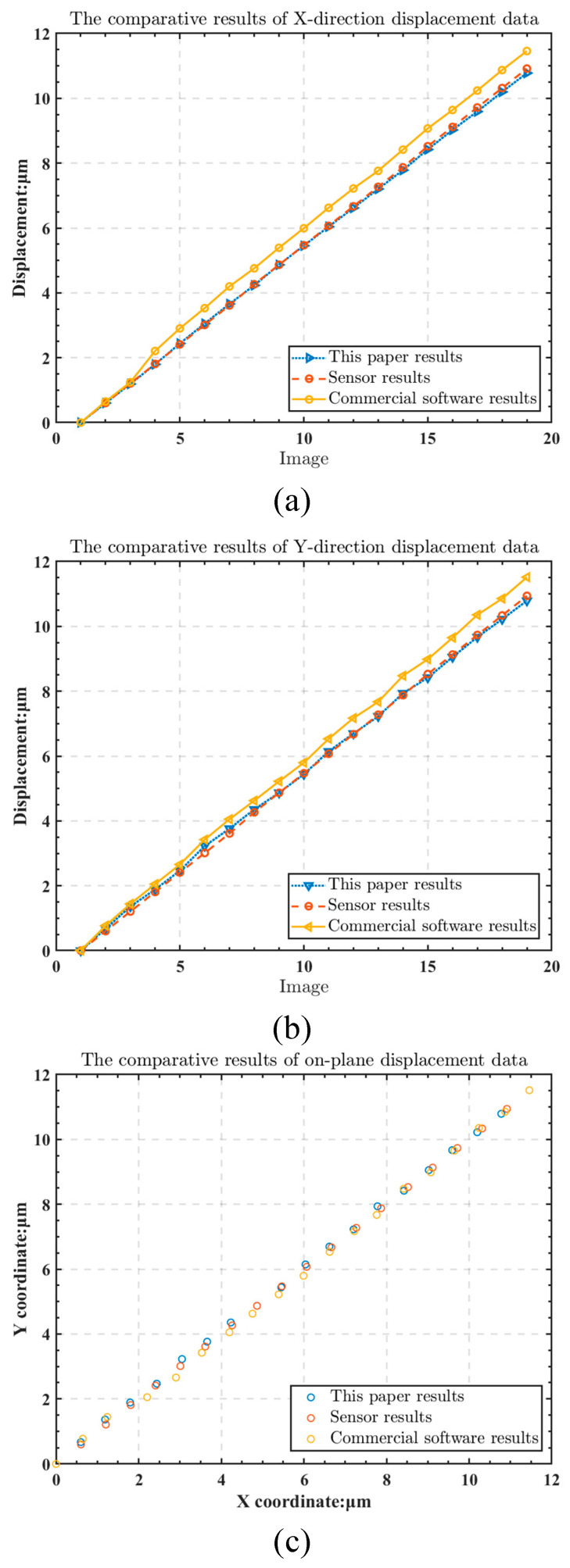
Experimental results: (**a**) the comparative results of X-direction displacement data; (**b**) the comparative results of Y-direction displacement data; (**c**) the comparative results of on-plane displacement data.

**Figure 14 micromachines-13-02159-f014:**
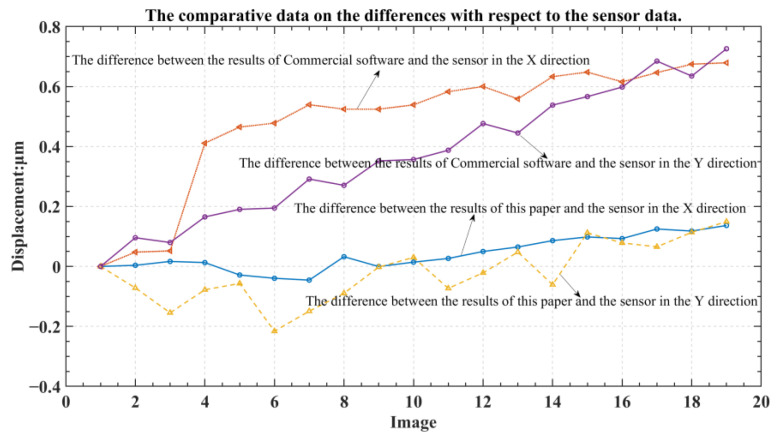
The comparative data on the differences with respect to the sensor data.

## Data Availability

The data presented in this study are available upon request from the corresponding author.

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
