# Peer review of "High-Precision Wafer Bonding Alignment Mark Using Moiré Fringes and Digital Grating"

_micromachines, 2022, doi:10.3390/mi13122159_

Round 1

Reviewer 1 Report

This paper provided a Moiré-based mark for high-precision wafer bonding alignment. And through practical experiments designed the rationality and practicability of the mark. Using grating marks and digital gratings based on moiré fringes can significantly improve the accuracy of calculating the deviation of the upper and lower wafers in the alignment process, and control the estimated error to within 10 nm. More improvements should be addressed as follows:

1.      In section 3.1:In recent years, Moiré fringes have been widely used to measure displacement, de- formation, and other fields.Could you provide some references on the measurements of displacement and deformation using Moiré fringes?

2.      The estimated error was finally controlled within 10 nm. What is the main reason for the current error between your results or current method with commercial calculation?

3.      For experimental verification, is the comparison of x-direction displacement data or y-direction displacement data the only standard for the error evaluation? Is there any other method or index?

Reviewer 2 Report

The authors need to address the below issues before publication.

1.       Please carefully review the manuscript and correct the wrong words, phrases or sentences before re-submission.

2.       Please combine figures 2, 3 and 4 together.

3.       Please combine figures 5 and 6 together.

4.       The title of 3.4 and 3.5 same?

5.       In figure 12(b), X direction should be Y direction?

6.       Please combine figures 16, 17 and 18 together.

7.       Error bars for figures 16, 17, 18 and 19.

8.       More references are required.

Round 2

Reviewer 2 Report

No further comments.